# High Throughput Manufacturing of Bio-Resorbable Micro-Porous Scaffolds Made of Poly(L-lactide-co-ε-caprolactone) by Micro-Extrusion for Soft Tissue Engineering Applications

**DOI:** 10.3390/polym12010034

**Published:** 2019-12-24

**Authors:** Xabier Mendibil, Rocío Ortiz, Virginia Sáenz de Viteri, Jone M. Ugartemendia, Jose-Ramon Sarasua, Iban Quintana

**Affiliations:** 1IK4-TEKNIKER, C/IñakiGoenaga 5, 20600 Eibar, Spain; xabier.mendibil@tekniker.es (X.M.);; 2Department of Mining-Metallurgy Engineering and Materials Science, School of Engineering, University of the Basque Country (EHU-UPV), 48013 Bilbao, Spainjr.sarasua@ehu.eus (J.-R.S.)

**Keywords:** micro-extrusion, bio-resorbable scaffold manufacturing, porous scaffolds, poly(L-lactide-co-ε-caprolactone), soft tissue engineering

## Abstract

Porous scaffolds made of elastomeric materials are of great interest for soft tissue engineering. Poly(L-lactide-co-*ε*-caprolactone) (PLCL) is a bio-resorbable elastomeric copolymer with tailorable properties, which make this material an appropriate candidate to be used as scaffold for vascular, tendon, and nerve healing applications. Here, extrusion was applied to produce porous scaffolds of PLCL, using NaCl particles as a leachable agent. The effects of the particle proportion and size on leaching performance, dimensional stability, mechanical properties, and ageing of the scaffolds were analyzed. The efficiency of the particle leaching and scaffold swelling when wet were observed to be dependent on the porogenerator proportion, while the secant moduli and ultimate tensile strengths were dependent on the pore size. Porosity, swelling, and mechanical properties of the extruded scaffolds were tailorable, varying with the proportion and size of porogenerator particles and showed similar values to human soft tissues like nerves and veins (E = 7–15 MPa, σ_u_ = 7 MPa). Up to 300-mm length micro-porous PLCL tube with 400-µm thickness wall was extruded, proving extrusion as a high-throughput manufacturing process to produce tubular elastomeric bio-resorbable porous scaffolds of unrestricted length with tunable mechanical properties.

## 1. Introduction

Elastomeric bio-resorbable scaffolds are of great interest for soft tissue engineering as they can be synthesized to show proper biocompatibility, low immunogenicity, and a proper resorption rate to degrade into products that can be eliminated from the host’s body [1,2,3]. The performance of those scaffolds can be tailored to provide suitable mechanical properties during tissue healing and regeneration [4,5,6] fostering the adhesion, proliferation, and differentiation of cells. Another usual requirement for scaffold manufacturing is porosity, which is needed to promote cell migration, nutrient transport, and waste removal through the material, while maintaining adequate mechanical properties. Numerous attempts to produce highly porous scaffolds by several techniques and different materials can be found in the literature [7,8,9]. Generally, pores of diameter greater than 100-µm are suitable for cell colonization and migration, while smaller diameters are intended to provide favorable physiological liquid and nutrient interchange [10,11].

In this regard, poly(L-lactide-co-*ε*-caprolactone) (PLCL) arises as a promising material, due to its tailorable mechanical properties, degradation rate [12,13,14,15,16,17], and its easy processing to manufacture scaffolds in which porosity can be produced and controlled [18,19,20,21,22,23,24]. The lactide to caprolactone proportion of the PLCL can be varied to match the aimed tissue properties for different clinical applications [25]. In particular, the PLCL with 70/30 L-lactide/ε-caprolactone proportion enhances the proliferation of Schwann cells and is commonly used for neural guide conduit fabrication for peripheral nerve healing [26,27,28]. Particle leaching is the most common technique to create porous scaffolds [29,30,31,32,33,34,35,36,37,38,39,40,41]. This technique allows for the production of scaffolds with well interconnected pores, whose characteristics are controlled by particle proportion, size, and distribution. Some of the advantages of this technique are that non-specialized equipment is needed, and that pore size and porosity can be effectively controlled by varying the size and quantity of leachable particles. The polymeric scaffold is immersed in water to dissolve the porogenerator particles, which generally are salt particles of sodium chloride (NaCl) or other soluble porogenerator such as sucrose or starch [30]. NaCl salt particles, contrary to other particle materials, show thermal stability in the range of polymer processing temperatures and good resistance to the organic solvents typically used in polymer handling and processing.

Most of the current porous scaffold manufacturing techniques are based on obtaining films [4,34,35] or more complex three dimensional (3D) geometries (like tubes) [6,18,23] by means of solvent exchange or evaporation, or more novel technologies like 3D printing [42,43] and electrospinning [44], which consist of material deposition. These methods allow for the production of a high degree of porosity, up to 90% [4,10,23,32,33,36], and are successfully used to prepare testing samples. However, they show low manufacturing throughput, limitations in size and resolution, and are difficult to implement in industrial processes. For example, Jeong et al. [18] used a cylindrical mold where, after introducing a mixture of sieved salt and PLCL solved in chloroform, the lumen of the tube was shaped by means of a home-made tool before a solvent was allowed to evaporate, obtaining a 6-mm outer diameter tube and around 20-mm length. This procedure is similar to the industrial indirect extrusion, where billets of material are pushed by a ram to obtain the desired profile, but the productivity of this process is limited by the length of the ram and material billet size. On the contrary, extrusion is a high throughput manufacturing method where a polymer is processed to obtain unrestricted-length 3D components characterized by a constant cross section, such as tubes, profiles, and films [37,38]. The micro-extrusion process is focused on the manufacturing of profiles and tubes containing dimensions and features in the sub millimeter range, which can be of great interest for tissue engineering of certain tissues as veins, nerves and tendons, where long micro-structured scaffolds are usually needed. In addition, the micro extruders, due to their inherent small size, use low amounts of material and show low material residence times, which makes this technique highly appropriate to process thermal sensitive materials, polymers under development, and expensive polymers.

There is very little informationregarding the extrusion of bio-resorbable polymers with the objective of manufacturing soft tissue healing scaffolds. For example, Salmoria et al. [45] successfully extruded tubes of 1.5 mm in diameter made of polycaprolactone loaded with silver particles, to be used as tubular guides to promote the self-regeneration of injured peripheral nerves. The tubes were not porous, and the flexural moduli of the obtained tubes were over 200 MPa, which is not appropriate for most soft tissues. Other authors extruded the polymer and the porogenerator to prepare a mixture, which was later processed by other means, to obtain the test samples. Washburn et al. [46] used the extruder to blend two immiscible polymers which were later hot pressed to obtain thick films. Further, Etxabide et al. [38] extruded fish gelatin mixed with lactose to promote its crosslink to subsequently inject the test samples.

Here, a simple particle leaching method was combined with micro-extrusion processing as a high throughput manufacturing technique to obtain porous bio-resorbable scaffolds made of PLCL. Samples of PLCL containing different salt particle sizes and proportions were micro-extruded to analyze the post leaching residual porogenerator amount, the geometric variation of the porous micro-extruded parts, their mechanical properties, and the polymer ageing. A micro-extruded porous PLCL tube with 400-µm wall thickness was produced in order to investigate the capabilities of this technique to manufacture porous tubes made of an elastomeric biodegradable material as scaffolds for vascular tissue engineering [19,23,24] or implants to promote the healing of tendons [39,40] and nerves [31,41,47].

## 2. Materials and Methods

### 2.1. Materials

Purasorb PLC 7015 (PLCL with 70/30 L-lactide/ε-caprolactone proportion copolymer) was supplied by Corbion Purac (Amsterdam, The Netherlands). The melting temperature of the polymer is 107.3–112.4 °C. The polymer was stored in sealed bags in a freezer at −20 °C.

Analysis grade Sodium Chloride (NaCl) salt provided by Scharlab (Barcelona, Spain) was used as porogenerator. The salt was milled in a blender, sieved using 25, 50, and 100-µm mesh sizes, and kept in silica-gel desiccator at room temperature (RT).

### 2.2. Methods

#### 2.2.1. Sample Preparation and Characterization

The extrusion of PLCL/NaCl was carried out using a Thermo Scientific Haake Minilab II micro extruder (Waltham, MA, USA). The processing parameters were constant for all the performed extrusions, applying a barrel temperature of 140 °C and a screw rotating speed of 10rpm. Neither cooling nor pulling wereused at the outlet during test sample extrusion. Only when extruding the final tube (which was flexible and fragile when warm) was a conveyor belt was placed next to the nozzle to facilitate handling.

Table 1 presents a description of the samples, particle sizes, and amount of porogenerator considered for the study, and their identifier numbers. This includes nine different mixtures, comprising the combination of 3 different salt/copolymer proportions (50/50, 60/40 and 70/30% in weight) and 3 different particle size ranges (0–50-µm, 25–50-µm and 50–100-µm). The outcome of particles smaller than 25-µm was negligible and non-viable for material preparation, therefore, the range including particles smaller than 25-µm (0–50-µm) was obtained mixing 50/50% in weight of 0–25-µm and 25–50-µm sieved salt to guarantee that at least the 50 wt.% of the particles were smaller than 25-µm. Particle size ranges and proportions were defined in preliminary extrusion tests whereby it was found that was not possible either to successfully extrude samples of polymer and NaCl mixtures containing porogenerator proportions higher than 80 wt.% with the defined experimental set. Prior to each extrusion, copolymer and sieved salt were weighed using a precision analytical balance from Mettler Toledo (Columbus, OH, USA). Subsequently, 14 g of each mixture were mechanically stirred and directly put in the micro-extruder feeder to manufacture 1400-mm length rods of each mixture with section of 1 × 4-mm, which were cut into 100-mm long pieces for handling purposes. The obtained salt/copolymer profiles, were stored in sealed bags in a desiccator at RT. From each mixture, 5 samples of 10-mm length (considering samples from the beginning, middle and ending sections of the extruded rod) were collected and prepared for characterization.

Dimensions of the extruded samples were measured by a 10-µm resolution digital gauge from Mitutoyo (Kawasaki-shi, Japan) and their mass was measured using an analytical balance from Mettler Toledo (Columbus, OH, USA). The samples were later rinsed in 50-mL of deionized water for 5 days in order to dissolve and leach the salt particles from the extruded rod. The water was refreshed every 24 h. The samples were dimensionally characterized right after the leaching process using the gauge to keep record of dimensional changes. They were dried at RT wrapped in blotter paper for 3 days. Then, the samples were put in a desiccator at RT for at least one week and, once the samples were completely dry, they were dimensionally controlled and weighted. Leached and non-leached samples were cryogenic cut in liquid nitrogen to inspect the breakage surface using scanning electron microscopy (FE-SEM ZEISS ULTRA plus Gemini, Oberkochen, Germany). Pore size was characterized using ImageJ software, measuring the Feret diameter of ellipsoids adjusted to the shape of not less than 50 pores for scaffolds with big particles, and up to 500 pores in the case of scaffolds made with the smallest particles.

#### 2.2.2. Mechanical Tests

Tensile test samples made of neat PLCL (without porogenerator) were prepared following the method applied by Ugartemendia et al. [48]. Porous scaffolds obtained from leached extruded parts were also used as samples for performing mechanical tests. These were performed by means of an Instron Model 3369K2004 (Norwood, MA, USA) applying the following test conditions: RT, 1KN, 5-mm/min, 50-mm clamp distance. Secant modulus was calculated at 2% of elongation.

#### 2.2.3. Thermal Tests

Thermal analysis of leached and non-leached samples stored in a desiccator at RT for 12 weeks were characterized by differential scanning calorimetry (DSC, Mettler Toledo GC 200 Star System, Columbus, OH, USA), applying a ramp from −60 °C to 180 °C at 20 °C/min.

## 3. Results

### 3.1. Leaching Performance

Figure 1a shows the differences observed in scaffold weight after the leaching process for the considered three particle size ranges. The weight losses were (42 ± 5%) for samples containing 50 wt.% salt proportion, (55 ± 2%) for samples with a 60 wt.% of salt, and (67 ± 3%) for samples having 70 wt.%. No weight loss differences were attributable to particle size when same salt proportion was considered. Figure 1b shows the sample weight loss deviation, which was calculated as the weight loss normalized to the theoretical maximum loss (i.e., 50%, 60% and 70%), showing the percental amount of porogenerator that remained in the leached sample. Samples containing 70 wt.% of salt showed a remnant of around (4 ± 5%), whereas for samples having 60 wt.% and 50% of salt, the remnant slightly increased up to (6 ± 5%) and (13 ± 8%) respectively.

Figure 2 shows SEM images of leached samples of each mixture. SEM images of pre and post leached samples are shown in Appendix A. Most of the non-leached samples showed uniform particle distribution with no aggregates and particle sizes inside the expected range for each mixture. Some samples showed the formation of a continuous polymer skin (without presence of particles or pores) on the top and bottom surfaces of the scaffolds. After leaching, there were no residual particles, as expected, due to the good leaching efficiency observed in weight loss analysis. Table 2 shows pore mean sizes measured by ImageJ in leached scaffolds, showing that the observed values were consistent with the porogenerator particle size used to prepare each scaffold.

Although there was a good correspondence between pore and particle sizes, SEM images show pores with elongated shapes along the scaffold length instead of the expected cube-shaped pores that should be formed after particle leaching.

### 3.2. Dimensional Variations

Figure 3 and Figure 4 show the percentage change in scaffold dimensions for wet (Figure 3) and dry (Figure 4) leached samples obtained using different salt weight proportions and particle sizes. The pre-leached samples originally measured 1 × 4 × 10-mm. The dimensional variations of the scaffolds in length, width, and thickness were plotted separately: change in scaffold length and width is shown in Figure 3a and Figure 4a, and change in scaffold thickness is shown in Figure 3b and Figure 4b, for wet and dry samples, respectively. Wet scaffolds showed an isotropic increase in their dimensions (swelling) for all particle sizes and proportions. The swelling for the most porous scaffolds produced with 70 wt.% of porogenerator was constant (50 ± 8%) regardless salt particle size, and the lowest compared to scaffolds with lower porosity. However, scaffolds produced with 60 wt.% of porogenerator showed a swelling of (63 ± 9%) for 0–50-µm particle size range, (88 ± 12%) for 25–50-µm particle size range and (105 ± 6%) for 50–100-µm particle size range, revealing an increase of swelling rate with the salt particle size. This did not happen to 50 wt.% samples, which showed an increase in swelling from small to medium particle size ranges ((109 ± 7%) for 0–50-µm and (121 ± 10%) for 25–50-µm), but a decrease for the 50–100-µm particle size range (74 ± 1.4%).

Unlike wet scaffolds, dry scaffolds showed an anisotropic change in their dimensions. Scaffolds with highest porosity (70 wt.% porogenerator) showed negligible length and width variation, while less porous scaffolds (50 wt.%) showed a maximal swelling of about (25 ± 9%) with a non-significant dependence on pore size. The change in scaffold thickness was similar and negative (about −30%) for all the measured samples, with no significant differences between scaffolds with different pore sizes and proportions.

### 3.3. Mechanical Properties

Table 3 shows the mechanical properties of porous scaffolds produced using different porogenerator sizes and proportions compared to neat PLCL. Samples containing 70 wt.% and 50 wt.% of particles in the 0–50-µm range showed secant modulus (E_2%_) values similar and even greater than the neat PLCL ((19 ± 4) and (11.9 ± 0.5) MPa compared to 12 MPa), while the rest of the samples showed lower E_2%_ values. All the porous samples showed ultimate strengths (σ_u_) and elongations at break (ε_u_) lower than the neat PLCL (maximums of (8.7 ± 1.5) MPa and (201 ± 9%) compared to 17 MPa and 441%). Values of E_2%_ and σ_u_ of samples of the same porosity were observed to decrease with the increase in pore size. Variations from (19 ± 4) to (6 ± 2) MPa, from (8.5 ± 0.6) to (2.8 ± 0.2) MPa and from (11.9 ± 0.5) to (7 ± 4) MPa were observed for values of E_2%_, and from (8.7 ± 1.5) to (4.7 ± 0.6) MPa, from (7.5 ± 2) to (3.1 ± 0.5) MPa and from (8.5 ± 0.4) to (4.2 ± 0.3) MPa in the case of σ_u_ values. The ultimate strength was similar for scaffolds with the same pore size but different porosity, indicating no influence of pore density in this mechanical property. The elongation at break (ε_u_) did not seem to follow a clear trend with the change in porosity nor dimensions, showing a nearly constant value of about 200%.

### 3.4. Thermal Analysis

Figure 5 shows the thermal behavior of leached (a) and non-leached (b) samples aged for 12 weeks at RT, considering different salt weight proportions and particle sizes. Leached samples appeared to be slightly crystalline with a cold crystallization peak (T_c_) at 60 °C, melting temperature at T_m_ = 100 °C, and glass transition temperature at T_g_ = 18.5 °C. All non-leached samples were amorphous, showing a single T_g_ at 23 °C (TGA (Appendix A) and TGA results (Appendix A) of each mixture are shown in Appendix A).

## 4. Discussion

Non-significant differences in weight loss due to different particle sizes were observed, which suggests that the efficiency of the leaching was not dependent on particle size but on initial porogenerator amount. Also, a modest but significant increase in weight loss deviation was observed when decreasing the salt proportion, with a maximum remnant salt value of approximately 13%. Considering the high standard deviation values, the measured remnants were close to that reported for the applied sample preparation method (<4 wt.%) [49] and no noticeable effect can be attributable to the presence of skin effect observed in some of the samples. Samples with low salt proportions contained non-porous copolymer volumes in the matrix where none salt was present (Figure 6a), leading to poorly connected salt particles, that eventually remained embedded in the copolymer matrix after the leaching process (Figure 6b). According to Reignier et al. [50] and Hou et al. [51], these isolated particles could be more difficult to leach from the polymeric matrix due to the decrease in contact points between particles. Particle volume fraction (φ_v_) was calculated from the weight fraction (φ_w_) and particle and polymeric matrix densities (ρ_p_ and ρ_m_, respectively) (see Appendix A for detailed information). Due to the difference between the PLCL and the NaCl particle densities (ρ_m_ = 1.22 g/cm^3^, from Corbion Purac safety data sheet and ρ_p_ = 2.165 g/cm^3^), there was a noticeable change between volumetric and weight fractions, e.g., for a porogenerator weight proportion of 50 wt.%, the fraction of total volume attributable to leachable particles is around 36 vol.%, which is far from the maximum packing fraction for cubic or even spherical particles (around 64 vol.% [52]). This means that some pores or particles could be not well connected to the porous network [50,51].

Wet porous PLCL scaffolds swelled isotropically and up to 100% for some of the considered pore density and size combinations. The swelling of the scaffolds could be caused either by the polymer matrix absorbing water, despite the low hydrophilicity of the PLCL (with a contact angle of 81°) [53], or by the osmotic pressure, where the polymer matrix acted as a membrane causing water penetration into the scaffold and prevented the free flow of the solution. Low pore density scaffolds showed higher swelling percentages than scaffolds with greater porosities. This could be due to the higher proportion of polymeric material liable to absorb water in those low porosity scaffolds, leading to a greater swelling than in scaffolds with higher porosities. However, scaffolds with the lowest pore density and biggest pore sizes (501050) did not follow this trend. The defects observed in particle distribution inside the polymeric matrix of this scaffold (Figure 6a) are detrimental to pore interconnectivity, which could cause a more restricted interchange with the leaching media. The non-porous volumes inside the copolymer matrix of this scaffold were in a more restrictive situation to get access to humidity and to freely swell than in the case of a uniform well-interconnected porous structure. Swelling ratios of scaffolds obtained with 50 and 60 wt.% porogenerator amounts were proportional to the particle size, but this did not occur in samples containing 70 wt.% salt. This fact could also be related to the scaffold pore amount and interconnectivity, since for higher proportions of porogenerator, greater homogeneity and pore interconnectivity can be achieved, and less polymeric material is liable to absorb humidity, eventually veiling the effect of particle size. These findings could be of great interest toscaffold design, since high porosities and implantation in wet state (physiological conditions) [31,54] are usually required.

Strong residual deformation anisotropy was observed in dry scaffolds, which showed length and width variations inversely proportional to their pore density, (as observed in wet conditions), but a negative thickness variation for all the scaffold preparation conditions (around −30%). The scaffolds that did not follow the observed trend in wet conditions (501050), once dry, exhibited similar length and width deformation (maximum of (25 ± 9%)) to scaffolds with different pore sizes but the same pore density (50 wt.%). This could be due to a lower recovery capacity of the deformed material due to the lesser elasticity of the observed non-porous volumes inside the matrix compared to the spongy porous structure. The measured negative thickness variation indicated the occurrence of a collapse during the drying process, which is also noticeable in the shape of the observed pores, which are elongated instead of cubic. The scaffolds were dried by placing them over blotter paper for three days before being put into a desiccator, producing the differential deformation measured in the thickness variation measurement and in the directional pore deformation observed in SEM cross section images of the scaffolds (Figure 7).

Tensile tests showed that the ultimate strength (σ_u_) and elongation (ε_u_) values of the porous scaffolds were lower than bulk PLCL, as expected. On the contrary, the secant modulus (E_2%_) of some scaffolds (005070 and 005050) yielded values comparable and even greater than the bulk material (12 MPa). This effect of material stiffening (increase of E_2%_ up to 60%) can be related to the manufacturing process, where the polymer was extruded with a remarkable quantity of rigid particles which eventually could cause a stretching and orientation of the material. All the tested scaffolds yielded similar values of elongation at break (ε_u_) around 200%, showing no clear trend related to neither porosity nor pore size. Pore size seemed to directly affect the resistance of the samples (in terms of ultimate strength and secant modulus), the samples with smaller pores being more resistant than those having bigger pores. The decrease in σ_u_ as the pore size increases, (strength loss of 46% for samples with 70 wt.%, 59% for samples with 60 wt.% and 50% for samples with 50 wt.%), could be due to a more heterogeneous structure of scaffolds with bigger pores, which makes them more prone to stress intensification. On the other hand, the ultimate strength of the scaffolds was observed to be independent of pore proportion, opposed to the results stated on literature, where the ultimate tensile strength is strongly dependent on the porosity degree [10,18,20,30,51,55,56], decreasing as the porosity increases. The discrepancy between our findings and the data reported in the literature could be attributable to the porosity rates considered in our study (50, 60 and 70 wt.%), which were lower than those stated in literature (up to 90 wt.%), and to the manufacturing process applied here, in contrast to currently applied laboratory techniques for film preparation. In addition, it is important to highlight that porous structure deformation and break mechanisms are complex, as reported by Gibson et al. [57], who commented on the different and diverse deformation mechanisms a foam can undergo (pore wall bending, pore face bending, collapse of pores, elastic buckling, brittle crushing etc.). Further studies must be carried out to investigate the effect of pores on the mechanical properties of extruded samples, e.g., compressive strength test, fracture toughness test, etc. Nevertheless, from the obtained results it can be inferred that mechanical properties of the obtained porous scaffolds were tailorable and similar to that of human soft tissues (human nerve E: (16 ± 2) MPa, σ_u_: (7 ± 0.6) MPa [31]; Skin E: 7.6 MPa, Cartilage E: 3 MPa [58]; Aortic valve E: (15 ± 6) MPa, Cerebral vein E: 7 MPa [59]).

Non-leached samples, aged for at least 12 weeks at RT, showed no crystallinity, being completely amorphous. The presence of NaCl particles seemed to directly affect the mobility and evolution of the PLCL chains, preventing the crystallization of the material. The presence of the particles in the matrix caused a delay in copolymer crystallization, similar to that reported by Wurm et al. [60] and Kiersnoswki et al. [61], who observed a retard in the crystallization of different polymers (PA6 and poly-*ε*-caprolactone) due to the presence of inorganic particles. On the contrary, leached samples underwent clear crystallization, revealing the characteristic T_g_ and crystallization peak of aged PLCL (70% LA 30% CL proportion) [12,35,48,62]. This crystallization behavior corresponds to the crystal form due to the L-lactide building blocks of the copolymer [13,63]. The presence of salt particles in the pre-leached samples and its age delaying effect should be considered for the storage and shelve life strategies of implantable products made by the presented method, as the PLCL evolves to crystalline structure once the salt is removed and could lead to variations in mechanical properties and resorption rates.

PLCL tubes with an inner diameter of 1.5mm, a wall thickness below 500-µm, and a length of 300-mm were extruded, proving the feasibility of the micro-extrusion process to manufacture porous elastomeric biodegradable scaffolds (Figure 8). This tube was extruded using NaCl particles of 0–50-µm size with a 70 wt.% proportion. To the best of our knowledge, this study reports for the first time the manufacturing of a microtube made of porous, biocompatible and bio-resorbable PLCL and manufactured by means of a continuous extrusion process. The production of porous tubular scaffolds made of bioresorbable polymers by means of high throughput manufacturing techniques can have an important impact in scaffold manufacturing as it allows for the production of several meters of product in a few hours. The validation of this kind of industrial production technique for scaffold manufacturing helps to narrow the gap between the laboratory and market, reducing the production costs and improving the repeatability and reliability of the manufactured products. Further experiments regarding in vitro cell culture tests such as cytotoxicity and cell viability assays and/or bio-mechano-reactor tests will be carried out to investigate the use of those micro-extruded scaffolds in soft tissue engineering, mimicking the conditions of the in vivo environment in the final application (cell type, culture media, blood flow conditions, etc.).

## 5. Conclusions

Porous scaffolds made of medical grade of Poly(L-lactide-co-*ε*-caprolactone) were manufactured by continuous extrusion using a twin-screw micro extruder. The copolymer was mixed with different proportions and particle sizes of NaCl, extruded, and subsequently leached in distilled water to obtain scaffolds with different porosity degrees and pore sizes. The leaching efficiency improved for higher porosities, being independent on the resultant pore size. Scaffold swelling was inversely affected by its porosity, i.e., lower porosities swelled more when wet. Once dry, only scaffolds which had highest porosities (70 wt.%) recovered their initial length and width values, while the rest of the scaffolds showed residual deformation. All the tested scaffolds exhibited thickness reduction once dry. Pore size and proportion seemed to influence scaffold mechanical properties, the scaffolds with smallest pores being the most resistant and stiff. More importantly, the secant modulus and ultimate tensile strength of the produced scaffolds were tailorable by varying the pore size and porosity degree, and similar to human soft tissues such as nerves, veins, or skin. This paper sets down for the first time the processing of medical grade PLCL by means of continuous extrusion to produce scaffolds with tailorable porosity and mechanical properties. A sub-millimeter walled porous tube up to 300-mm in length was extruded, proving the capability of the micro-extrusion process as a high-throughput manufacturing technique for the production of elastomeric biodegradable scaffolds appropriate for the reconstruction and healing of nerves, veins, and other soft tissues.

## Figures and Tables

**Figure 1 polymers-12-00034-f001:**
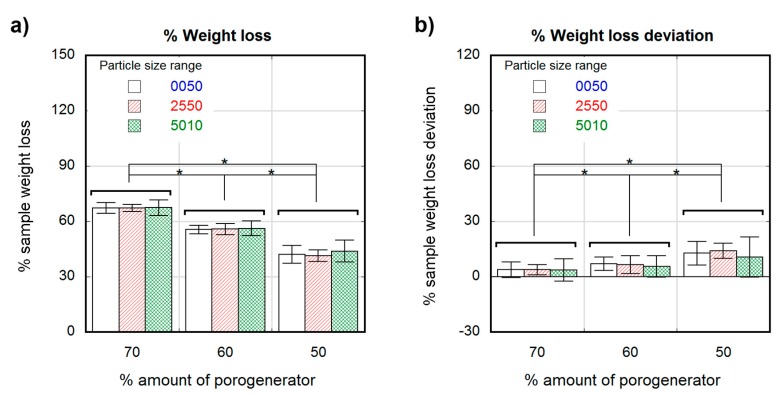
Percental weight loss after leaching of samples containing different salt particle sizes (white 0–50-µm, stripe pattern 25–50-µm and cross pattern 50–100-µm) as a function of the theoretical percental amount of porogenerator (**a**); Percental deviation from theoretical to experimental weight loss of samples containing different particle sizes (white 0–50-µm, stripe pattern 25–50-µm and cross pattern 50–100-µm) as a function of the theoretical percental amount of porogenerator (**b**). Significance level according to the Student *t*-test * *p* < 0.05.

**Figure 2 polymers-12-00034-f002:**
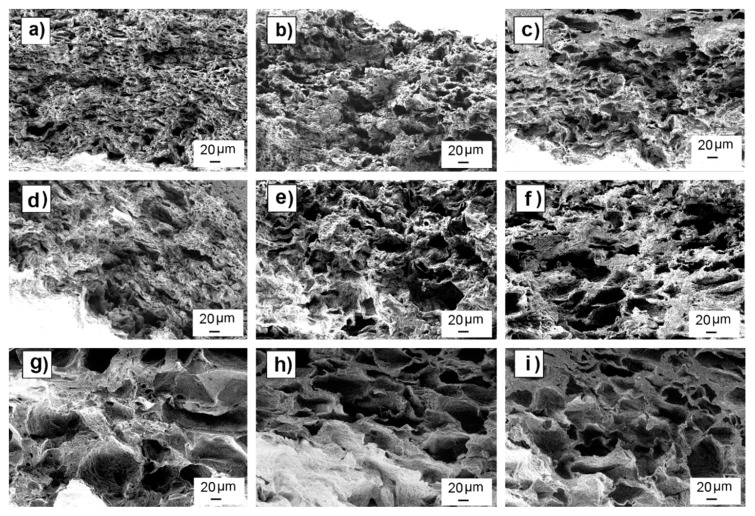
SEM images of nine different leached scaffolds arranged by their particle size range (row) and porogenerator weight proportion amount (column). Scaffolds made with particles in 0–50-µm range and proportion of 50 wt.% (**a**); 60 wt.% (**b**); and 70 wt.% (**c**); scaffolds made with particles in 25–50-µm range and proportion of 50 wt.% (**d**); 60 wt.% (**e**); 70 wt.% (**f**); scaffolds made with particles in 50–100-µm range and proportion of 50 wt.% (**g**); 60 wt.% (**h**), and 70 wt.% (**i**).

**Figure 3 polymers-12-00034-f003:**
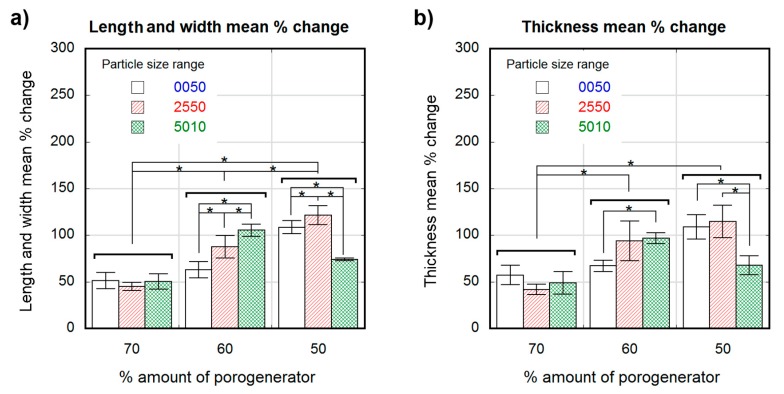
Dimensional variation in percental values for wet scaffolds obtained from different particle sizes (white 0–50-µm, stripe pattern 25–50-µm, cross pattern 50–100-µm) and proportions as a function of the theoretical percental amount of porogenerator: Scaffold length and width mean % change (**a**); scaffold thickness mean % change (**b**). Significance level according to the Student *t*-test * *p* < 0.05.

**Figure 4 polymers-12-00034-f004:**
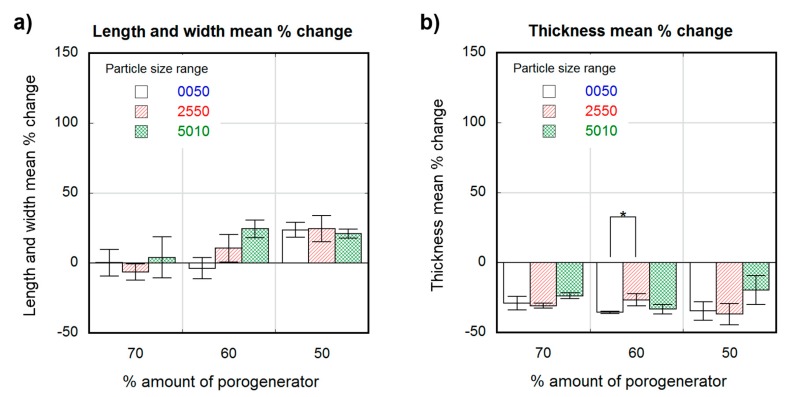
Dimensional variation in percentagevalues for dry scaffolds obtained from different particle sizes (white 0–50-µm, stripe pattern 25–50-µm, cross pattern 50–100-µm) and proportions as a function of the theoretical percental amount of porogenerator: scaffold length and width mean % change (**a**); scaffold thickness mean % change (**b**). Significance level according to the Student *t*-test * *p* < 0.05.

**Figure 5 polymers-12-00034-f005:**
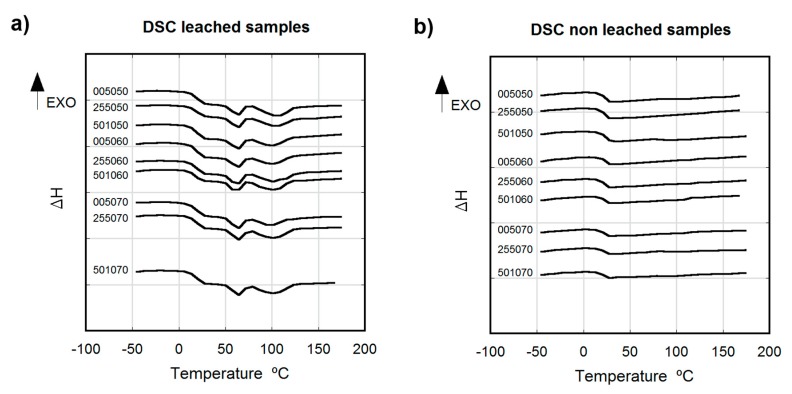
Thermograms of leached (**a**) and non-leached (**b**) aged samples (12 weeks at RT) produced using different porogenerator proportions and particles sizes (exothermic behavior upwards).

**Figure 6 polymers-12-00034-f006:**
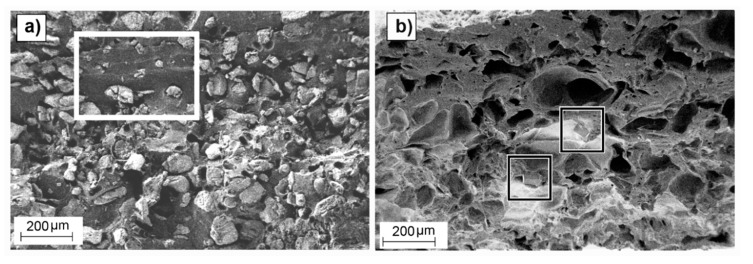
SEM images of cross-section surfaces of 501050 non-leached samples (**a**) and leached samples (**b**) showing non-porous polymer regions (outlined by a white rectangle in (**a**)) and remaining salt particles (outlined by squares in (**b**)).

**Figure 7 polymers-12-00034-f007:**
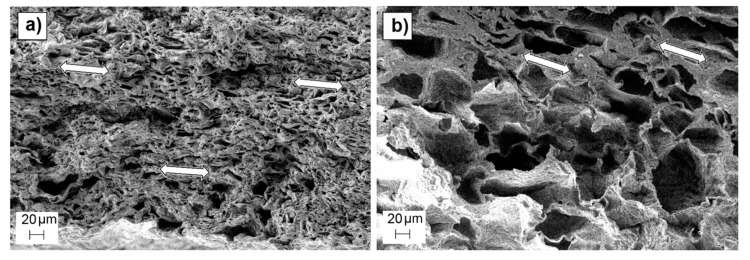
SEM images of cross-section surfaces of 005050 (**a**) and 501070 (**b**) scaffolds, showing the observed directional pore deformation highlighted by white arrows.

**Figure 8 polymers-12-00034-f008:**
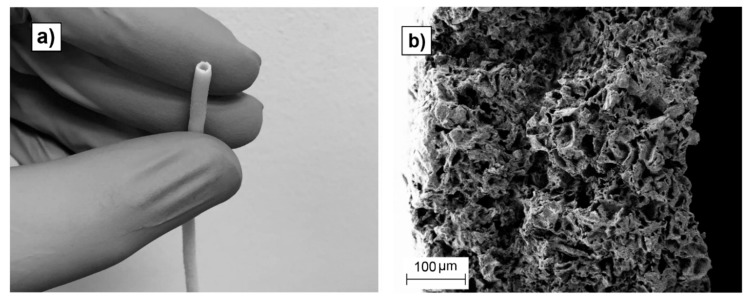
Tube with an inner diameter of 1.5-mm, a wall thickness of 400-µm, and a length of 300-mm produced by micro-extrusion (**a**); SEM image of a section of the tube wall (<500-µm) showing its porous structure (**b**).

**Table 1 polymers-12-00034-t001:** Identifier numbers of the considered porogenerator amounts and particle size ranges. Mixtures containing 50%, 60%, and 70% in weight of salt particles in 0–50-µm, 25–50-µm and 50–100-µm size ranges were prepared.

Amount of Porogenerator wt.%	Particle Size Range µm		
	00–50	25–50	50–100
**70**	005070	255070	501070
**60**	005060	255060	501060
**50**	005050	255050	501050

**Table 2 polymers-12-00034-t002:** Mean values and standard deviation of Feret diameter of ellipsoids adjusted to the pore shapes in leached samples with 50%, 60% and 70 wt.% of pore densities and particle sizes in 0–50, 25–50, and 50–100-µm ranges.

Amount of Porogenerator wt.%	Particle Size Range µm		
	00–50	25–50	50–100
**70**	20 ± 10	30 ± 20	50 ± 30
**60**	20 ± 10	30 ± 20	50 ± 20
**50**	12 ± 8	20 ± 20	60 ± 40

**Table 3 polymers-12-00034-t003:** Secant modulus E_2%_ (MPa), ultimate strength σ_u_ (MPa) and elongation at break ε_u_ (%), measured by tensile tests, of porous scaffolds produced using 50, 60 and 70 wt.% of porogenerator and particle sizes in 0–50, 25–50 and 50–100 µm size ranges, and compared to neat PLCL.

Porogenerator Proportion (wt.%)	Sample	Material Property		
		E_2%_ (MPa)	σ_u_ (MPa)	ε_u_ (%)
**70**	005070	19 ± 4	8.7 ± 1.5	187 ± 6
	255070	9.7 ± 1.1	7.2 ± 0.8	218 ± 7
	501070	6 ± 2	4.7 ± 0.6	215 ± 20
**60**	005060	8.5 ± 0.6	7.5 ± 2	204 ± 3
	255060	6.4 ± 1.4	5.7 ± 1	190 ± 20
	501060	2.8 ± 0.2	3.1 ± 0.5	180 ± 20
**50**	005050	11.9 ± 0.5	8.5 ± 0.4	201 ± 9
	255050	10 ± 1	6.7 ± 0.8	190 ± 10
	501050	7 ± 4	4.2 ± 0.3	150 ± 30
**0**	PLCL	12	17	441

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
