# Peer review of "High Throughput Manufacturing of Bio-Resorbable Micro-Porous Scaffolds Made of Poly(L-lactide-co-ε-caprolactone) by Micro-Extrusion for Soft Tissue Engineering Applications"

_polymers, 2019, doi:10.3390/polym12010034_

Round 1

Reviewer 1 Report

After reviewing the manuscript High throughput manufacturing of bio-resorbable micro-porous scaffolds made of poly(L-lactide-co-ε-caprolactone) by micro-extrusion for soft tissue engineering applications.

The manuscript is well designed and the methodology adequate. We’ll recommend a minor revision. However, Some clarifications need to be made before manuscript acceptance.

Abstract, page 1 line 23. Remove soft tissue like nerves and veins, these can’t be named as soft tissues.

Introduction.

High porosity values and high pores interconnectivity are often recommended for tissue engineering procedures. Several attempt have been made to reach high porosity values, maintaining adequate mechanical properties and this should be explained in the introduction.

These references could be helpful and should be cited in addition to ref 32 and 45.

Porosity:

-Karageorgiou V, Kaplan D. Porosity of 3D biomaterial scaffolds and osteogenesis. Biomaterials. 2005 Sep;26(27):5474-91.

-Hu J, Sun X, Ma H, Xie C, Chen YE, Ma PX. Porous nanofibrous PLLA scaffolds for vascular tissue engineering. Biomaterials. 2010 Nov;31(31):7971-7.

-Cho YS, Hong MW, Quan M, Kim SY, Lee SH, Lee SJ, Kim YY, Cho YS. Assessments for bone regeneration using the polycaprolactone SLUP (salt-leaching using powder) scaffold. J Biomed Mater Res A 2017;12:3432-3444.

-Gandolfi MG, Zamparini F, Degli Esposti M, Chiellini F, Fava F, Fabbri P, Taddei P, Prati C. Highly porous polycaprolactone scaffolds doped with calcium silicate and dicalcium phosphate dihydrate designed for bone regeneration. Mater Sci Eng C Mater Biol Appl. 2019 Sep;102:341-361.

Page 2 line 92 Author should better explain the potential clinical application of the tested scaffold. Soft tissue engineering is too vague, especially if they referred this term to tendon, skin, vascular network, or even nerves. All these tissues are different, therefore the precise application is crucial. Please explain and modify in the text.

Materials and Methods:

Table 1, and  generally all figures. The definition of samples used in the present study is a bit confusing (e.g 005070). I suggest to hyphenate the samples tested (e.g 00-50-70)

Moreover, I suggest to add a 0 to the 50-100 formulation (e.g50-100-70). In this way the reader may immediately recognize the sample and percentages authors are referring to.

Results:

Page 5 line 178, What “skin  formation “do author refer? Please explain.

It could  be interesting to assess Pores interconnectivity on leached samples. A recent study investigated filler distribution and pores interconnectivity using Micro-CT on polymer based scaffolds (Tatullo et al 2019). Could it be possible to use a similar protocol to assess pores interconnectivity? If is not possible, this could be a limitation of the study.

-Tatullo M, Spagnuolo G, Codispoti B, Zamparini F, Zhang A, Esposti MD, Aparicio C, Rengo C, Nuzzolese M, Manzoli L, Fava F, Prati C, Fabbri P, Gandolfi  MG. PLA-Based Mineral-Doped Scaffolds Seeded with Human Periapical Cyst-Derived MSCs: A Promising Tool for Regenerative Healing in Dentistry. Materials (Basel)2019 Feb 16;12(4).

Discussion

Other studies used different porogens, including gelatin microspheres, paraffin microspheres, or sugar crystals in order to get a more homogeneous structural pores. What advantages may the present technique provide? These paper should be discussed

-Draghi L, Resta S, Pirozzolo MG, Tanzi MC. Microspheres leaching for scaffold  porosity control. J Mater Sci Mater Med. 2005 Dec;16(12):1093-7.

-Tan Q, Li S, Ren J, Chen C. Fabrication of porous scaffolds with a controllable microstructure and mechanical properties by porogen fusion technique. Int J Mol Sci. 2011 Jan 25;12(2):890-904

What degradation rate  authors expect for this scaffold? Perhaps this should be implemented in the discussion as could directly influence the potential clinical application of this scaffold. Could author provide some studies which investigated PLCL degradation rate in SBF?

Reviewer 2 Report

This manuscript fabricated a poly(L-lactide-co-ε-caprolactone) (PLCL) porous scaffolds via extrusion and particle leaching. By changing the salt particle sizes and proportions, the mechanical properties and porosity could be varied. This manuscript may be recommended to be published in this journal. However, there are some issues needed to be addressed:

Page3: As the authors said, it was not possible to extrude polymer containing porogenerator proportions higher than 80% wt but there was already a study reported that the solid-state extrusion could successfully extrude scaffolds with 85% of NaCl. As a consequence, the statement should be more accurate. The data summarized in Table 2 and Table 3 may be presented in a column to make it much clearer. TGA was conducted but there was no discussion about the result. If you mentioned TGA in your methods part, the results should be described and maybe be discussed. What is the purpose of DSC test? Why the aging delaying effect should be considered or what is its relation to storage and shelve life strategies? Please give a detailed statement. Fig 7. Please add the mark to distinguish two figures. Since the scaffolds are made for tissue engineering applications, in vitro test is suggested to access the cytotoxicity. There are some reports to prepare porous scaffolds using salt leaching. The authors are suggested to introduce and discuss them in the revision, such as, Polymers 2016, 8, 213-225; Journal of Biomedical Materials Research, Part A 2019, 107A, 654–662; ACS Biomaterials Science & Engineering 2019, 5, 2998-3006

Reviewer 3 Report

Line 57 - Replace 'in' with 'of'

Line 59 - Do you have any reference to support your statement that 3D printing demonstrated low manufacturing throughput?

Line 120 - Replace 'weighted' with 'weighed'

Line 130 - same comment as above

I figure 3 and figure 4, spelling of length is incorrect

Round 2

Reviewer 1 Report

The manuscript has been improved and it can be accepted in this form.

Reviewer 2 Report

The authors answered my concerns. Now it is recommanded to be published in this journal.